# Forest Cover Change and Climate Variation in Subtropical Chir Pine Forests of Murree through GIS

Lubna Ansari [1],*, Waqar Ahmad [1], Aamir Saleem [1], Muhammad Imran [2], Khafsa Malik [3], Iftikhar Hussain [1], Hina Tariq [1] and Mubashrah Munir [4]

1 Department of Forestry and Range Management, Pir Mehr Ali Shah (PMAS) Arid Agriculture University, Rawalpindi 46000, Pakistan

2 Institute of Geo-Information and Earth Observation, Pir Mehr Ali Shah (PMAS) Arid Agriculture University, Rawalpindi 46000, Pakistan

3 Department of Botany, Pir Mehr Ali Shah (PMAS) Arid Agriculture University, Rawalpindi 46000, Pakistan

4 Department of Botany, University of Veterinary and Animal Sciences (UVAS—Pattoki Campus), Lahore 54000, Pakistan

* Correspondence: lubna.ansari@uaar.edu.pk

**Abstract:** Forests are valuable natural resources, beneficial for the storage of carbon, production of oxygen, protection of soil and controlling the water cycle. Despite forests providing different services to the environment, they are being destroyed at an alarming pace. Forest cover change in Murree, Pakistan over the past few years has created different climatic issues. There was a research gap on the detection of forest cover change along with climate variation in the past few years in Murree, so there is a dire need to highlight the above problem in the respective site. Further, it was equally important to keep an eye on the drivers of deforestation to give or suggest solutions accordingly to curb deforestation. The main objectives of this study are to assess forest cover change in subtropical Chir pine forests in Murree, Pakistan over the last 20 years (2001–2021) and to correlate forest cover change with the climatic variables (minimum and maximum temperature and precipitation) of the study area during this time span (2001–2021). This research also intends to identify the main drivers of deforestation in the study area. Five land-use land-cover (LULC) categories are demarcated and classified by applying a supervised classification technique (MLC) through GIS. The accuracy of classified images is assessed and analyzed using KAPPA analysis for the agreement of the image classification. Climatic data are interpolated by empirical Bayesian kriging (EBK) interpolation and it was correlated with forest cover change graphically. Drivers of deforestation are identified through a questionnaire and analyzed in SPSS. The results showed that forest area has decreased 8.26% in Murree from 2001 to 2021. Fuelwood (54%), agriculture expansion (22%), timber production (16%), and urbanization (8%) are recorded as drivers of deforestation in the study area. Climatic variables (maximum and minimum temperature and precipitation) had also shown variation in Murree, as the average maximum temperature has risen 0.26 °C, the average minimum temperature has risen 1.71 °C and annual rainfall has decreased 139.8 mm in the past 20 years (2001–2021), showing that forest decline has caused an increase in temperature and a decrease in rainfall in Murree.

**Keywords:** deforestation; empirical bayesian kriging; land-use land-cover; subtropical Chir pine





## 1. Introduction

Forests are valuable natural resources given to humankind as a gift from Allah Almighty. These natural resources would be useless if we do not maintain and use them sustainably [1]. Trees are beneficial for the storage of carbon, production of oxygen, protection of soil and controlling the water cycle. They maintain the human as well as the natural food systems and provide habitat to numerous animals, including human, providing materials for construction purposes. Forests (trees) are the best air cleaners, and because of the essential role of forests and trees in the terrestrial environment, the existence of various

animals, including humans, on Earth is difficult without forests [2]. Despite all that forests do for the environment, they are being destroyed at an alarming pace. Deforestation claims 46–58 thousand square miles of forest per year in the world [3]. (Adams, 2012). Worldwide, forest loss outnumbers forest production, but the aspects and magnitude of change vary depending on area and driving forces [4].

Significant deforestation has led to a rapid transition in forest and land-cover over the last few decades, especially in tropical forests. The main drivers of the transitions are agricultural intensification, rural settlements and urbanization, which are all aimed at meeting the rising demand of an ever-increasing population [5].

Deforestation is also affecting the climate as changes in forest cover have an impact on the local climate through modifying energy and water exchanges between the land and the atmosphere [6]. During the time span of 2001–2020 global surface temperature was 0.99 (0.84–1.10) °C higher than 1850–1900, with more increase over land (1.59 [from 1.34 to 1.83] °C) than over oceans (0.88 [from 0.68 to 1.01] °C) [7]. The increasing temperature of the Earth is a consequence of the massive emissions of greenhouse gases into the atmosphere, which is directly proportional to tree cutting [8]. Deforestation has been identified as the primary cause of a decrease in rainfall and an increase in temperatures globally [9].

Humans on the earth's surface have not significantly altered a few landscapes. Agriculture, mining, deforestation and development all have an impact on changing land-use patterns, which is a major source of ongoing environmental issues, as land-use and land-cover change (LULCC) is highlighted as an important factor in triggering environmental degradation [10]. Therefore, for observing global and regional changes in the environment, recording forest cover and vegetation is important.

LULCC is the term used around the world to describe how human activities affect the earth's terrestrial surface. While people have been altering land for many years to obtain livelihoods and other necessities, the strength and level of LULCC have scaled up now more than they were in the past. These differences are causing tremendous changes in ecosystems and environmental processes at the local, regional and global levels. As a result, LULCC has an important role in studying and reviewing today's global modified scenarios, as the availability of LULCC information is critical to providing constructive criticism for the management and decision-making planning of environmental issues [11–14].

This LULCC pattern is altering and transforming forest cover, biodiversity, agriculture and land productivity [15], and these LULCC patterns in the areas might cause climatic drivers such as rainfall and temperature to shift [16]. Many studies have been published around the world that use LULCC to determine the impact of climate change on forests. Changes in LULC caused by any factor (natural or anthropogenic) have a significant impact on global and regional scale patterns, which in turn influence weather and climate [17]. Therefore, the detection of LULC changes is being identified as a critical research factor for global environmental changes [18].

For environmental analysis and management, a unified remote sensing and Geographical Information Systems (GIS) approach has gained popularity due to its capacity for handling and controlling satellite and geographic information to meet the growing demands of planners and policymakers for precise and reliable LULCC information [19]. RS and GIS techniques are mostly used to assess land-use/land-cover and its effects on environmental hazards [20,21].

This technique offers an exclusive historical database for analyzing land-cover changes over time [22]. Various types of land-cover change studies have been conducted using satellite imagery, including deforestation assessments, crop stress identification and a variety of other environmental studies [23–26]. Environmental shifts can be detected using climatic factors such as temperature and precipitation and the effects of this detection can then be used for environmental sustainability processes [5].

In Pakistan, forests have long been known as a national treasure. The forests of Pakistan reflect the enormous climatic and physiographic variation and are continuously under pressure. The natural forests of Pakistan are being deforested at an alarming rate.

Reducing Emissions from Deforestation and Forest Degradation (REDD+) [27] recorded deforestation as being about 27,000 ha annually in Pakistan's Himalayan pine forest.

During the past couple of decades, Pakistan has lost a considerable number of forests. The main reason or driver of deforestation in Pakistan is the increasing demand for timber and fuelwood from the growing population [28]. This deforestation has caused climate variation as the rainfall pattern has also changed as a result of forest degradation [10].

The remaining forests in Pakistan are both fragile and special. They are vital for the ecological services they offer to society [29]. With the prevailing depletion of forest areas worldwide, it is vital that we handle these renewable resources sustainably. Therefore, it is necessary to have solid information about the LULC in order to develop and implement effective forest management policies and practices [10].

The current study area is the sub-tropical Chir pine forest, Murree. This area is a mountainous region, and mountain forests offer numerous benefits to both upstream and downstream communities, most importantly the sustainability of watersheds and transportation networks. They are also valuable as biodiversity hotspots, producers of timber, fuelwood and non-wood products, tourist and leisure destinations, and sacred sites. Furthermore, they are being seen as potential carbon sinks to help offset climate change [30]. However, these Himalayan forests are under great pressure due to illegal logging, urbanization and agricultural expansion [31]. LULC change is becoming a major issue in these Himalayan forests of Pakistan [32], as LULC change is most noticeable in the subtropical and moist temperate forest regions [33]. Therefore, these LULCC data are important to utilize in quantifying forest cover change and measuring the influence of forest cover change on rainfall and temperature rise in these areas. Research on detecting forest cover change and correlating it with climate variables was missing, and little was known about the drivers of deforestation in Murree, Pakistan. Classifying satellite imageries to detect forest cover change and correlating with climate variables and understanding drivers of deforestation in the mountainous region such as Murree was highly important in the climate changing scenarios.

This research intended to assess the change in forest cover in the study area from 2001 to 2021 and to correlate the climatic variables (minimum and maximum temperature and precipitation) with forest cover change. This research also identified drivers of deforestation in Murree, Pakistan.

## 2. Materials and Methods

### 2.1. Study Area

Murree is a tehsil in district Rawalpindi, Pakistan. It is the natural area of a Chir Pine forest, which is maintained using shelter-wood silviculture system. Murree Forest Division (MFD) is a 47,285-acre forest land under the administration of Punjab Forest Department. It serves as a significant sub-watershed of the Indus and Jehlum River systems.

The study site's latitude and longitude are 33 degrees 47′15″ to 33 degrees 54′47″ N and 73 degrees 16′54″ to 73 degrees 29′18″ E. Elevation of the forest ranges from 940 to 1873 m from sea level, and the study area's temperature ranges from −5 degree centigrade in winter to 40 degrees centigrade in summer. The average annual rainfall is approximately 1140 mm per year [34].

*Pinus roxburghii* (Chir Pine) is the principal tree species in the study area. Other main tree species are *Quercus incana* (rhin), *Pyrus pashia* (batangi), *Pinus wallichiana* (kail), and the associated understory flora consists of *Dodonea viscosa* (sanatha), *Capparis decidua* (karir), *Adhatoda vasica* (Bahekar), *Cannabus sativa* (Bang), *Berberis lycium* spp. (sumblu). Figure 1 is showing map of tehsil Murree, district Rawalpindi, Pakistan.

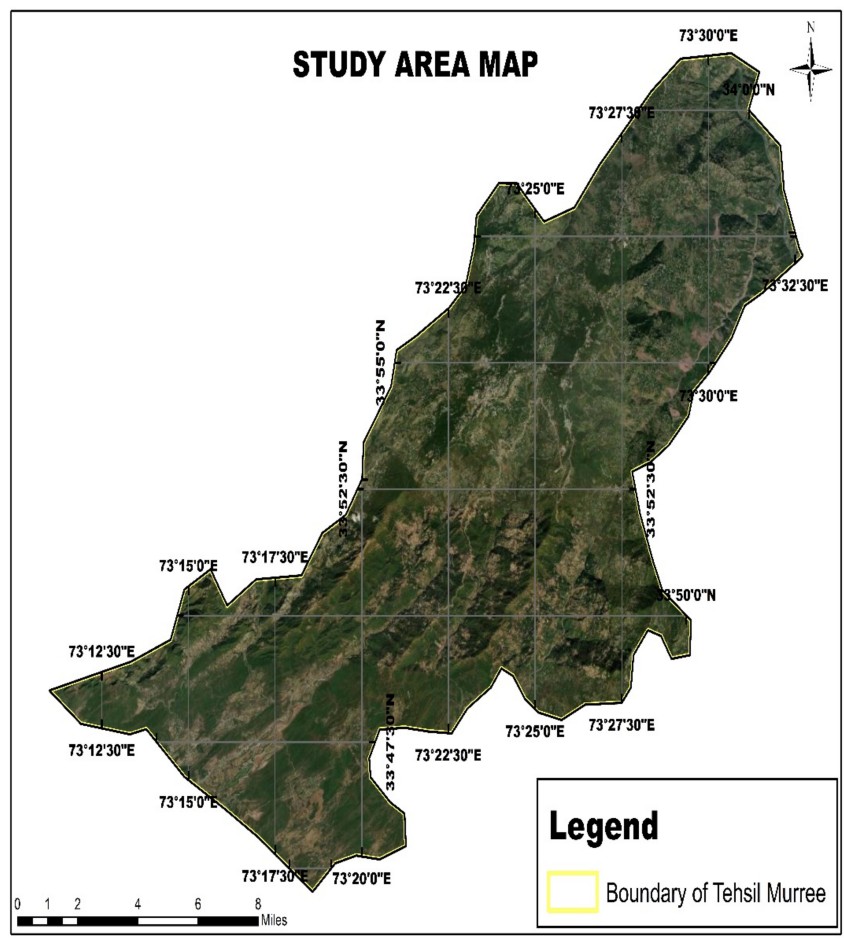

**Figure 1.** Map of Tehsil Murree.

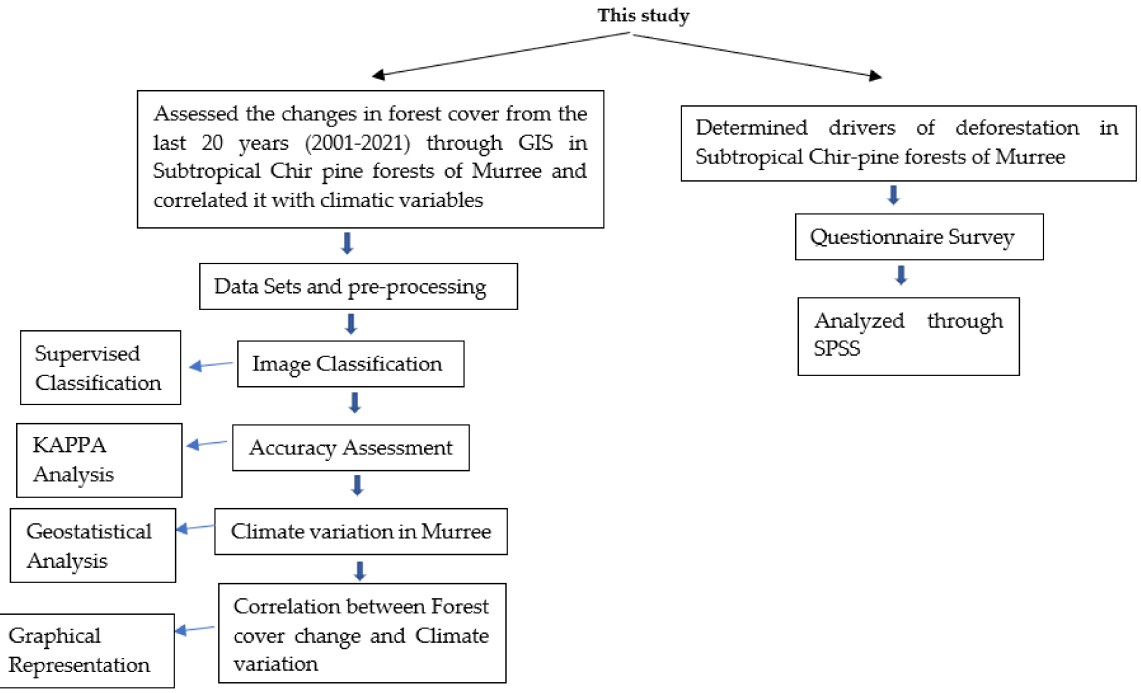

## 2.2. Study Design

The current study used supervised classification to find out the changes in forest cover from the last 20 years (2001–2021) through GIS. This is the most commonly adopted classification procedure [35]. Questionnaire survey was conducted to find out drivers of deforestation in the study area [36].

## 2.3. Data Sets and Pre-Processing

To detect forest cover changes in study area, data was gathered from remotely sensed satellite data (Landsat images), obtained from USGS website. Three satellite images for the years 2001, 2011 and 2021 were used to examine and analyze forest cover change in study area. The reason for this 10-year gap was the SLC-off data in Landsat 7 ETM, which shows data gaps after 31 May 2003 [37]. Secondly, because deforestation is a slow process, it needs to have a time lag to detect changes. In order to avoid the seasonal variations all satellite images were selected from the same season [8]. The images were of the same spatial resolution of 30 m to make them suitable for comparison of changes that occur in the time under contemplation (Table 1). Climatic data were gathered from Pakistan Meteorological Department. Questionnaire survey was carried out to collect primary data on the drivers of deforestation in the study area. In total, 10 union councils in Murree were selected and 5 respondents from each UC were interviewed randomly. Total 50 respondents were interviewed from the study area.

**Table 1.** Characteristics of Satellite images.

| Sensor | Acquisition Time | Path and Row | Spatial Resolution |
|---|---|---|---|
| Landsat 5 TM | 29 September 2001 | 150/036 | 30 m |
| Landsat 5 TM | 25 September 2011 | 150/036, 150/037 | 30 m |
| Landsat 8 OLI | 19 August 2021 | 150/036 | 30 m |

The first step of image analysis was image pre-processing in which atmospheric and topographic correction, layer stacking were applied to datasets. As the study area was not coming completely in single tile for the year 2011, therefore mosaicking was performed for the year 2011 and two images of path/row 150/036 and 150/037 were mosaicked to fix the issue. After mosaicking, study area was extracted with the help of shapefile.

## 2.4. Method of Image Classification

Before image classification, keeping in view the aim and scope of study, a simple, comprehensive and appropriate classification scheme was developed to examine and monitor the forest change in Murree since 2001. Five LULC classes, namely, forest land, cropland, wetlands, settlements and other lands (barren) was identified in satellite images.

LULC classes were produced by supervised classification of the satellite imagery. Images from the study region were taken through three different stages to generate the study area's land-cover classes. These include (1) collection of training data; (2) selection of training sample (signatures); (3) selection of appropriate approaches to classification. The following resulting 5 LULC classes were recognized and mapped: forest land, cropland, wetlands, settlements and other lands (barren). Figure 2 is showing google earth and GPS points for the years 2001, 2011 and 2021, used for classification of Landsat imagery.

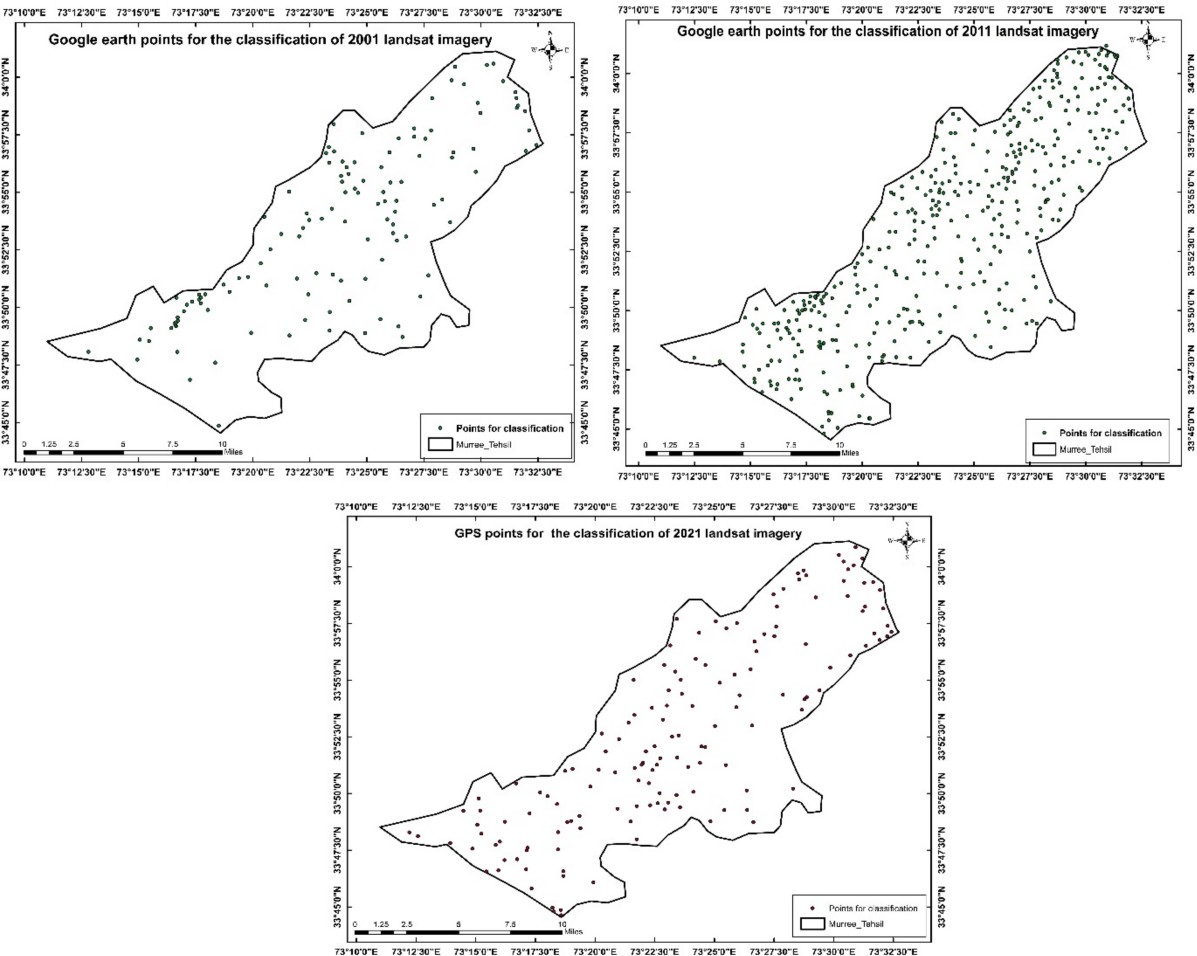

**Figure 2.** Map showing Google Earth and GPS points for classification of 2001, 2011 and 2021 Landsat imagery.

### 2.5. Accuracy Assessment of the LULC Classification

Accuracy assessment of the LULC classification was performed. The satellite and GPS-based reference points were overlaid with the classification image and the corresponding classified points to calculate the error matrix. Figure 3 is showing reference points, for the years 2001, 2011 and 2021, used for accuracy assessment.

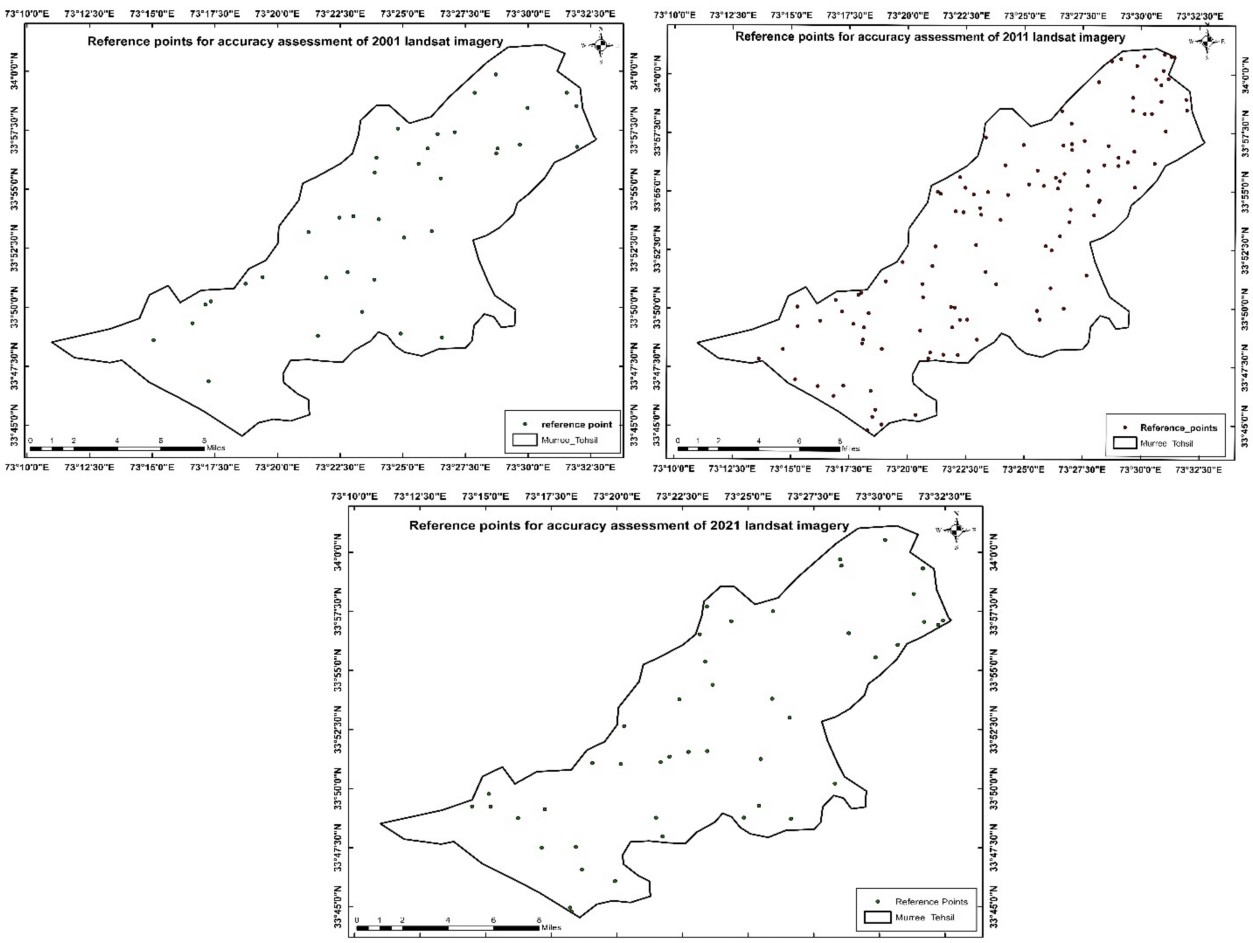

**Figure 3.** Map showing reference points for accuracy assessment of 2001, 2011 and 2021 Landsat imagery.

*2.6. Error Matrix*

Error matrix of the years 2001, 2011 and 2021 for the reference points and the classified points is shown in Tables 2–4 respectively.

**Table 2.** Error matrix of 5 classes of 2001.

| Classified | Forest Land | Settlements | Wet Land | Barren Land | Crop Land | Total Reference Points |
|---|---|---|---|---|---|---|
| Forest Land | 7 | 0 | 1 | 0 | 0 | 8 |
| Settlements | 2 | 10 | 1 | 1 | 0 | 14 |
| Wetland | 0 | 2 | 5 | 1 | 0 | 8 |
| Barren Land | 0 | 0 | 1 | 4 | 0 | 5 |
| Cropland | 1 | 2 | 0 | 0 | 5 | 8 |
| Total Classified Points | 10 | 14 | 8 | 6 | 5 | 43 |

**Table 3.** Error matrix of 5 classes of 2011.

| Classified | Forest Land | Settlements | Wet land | Barren Land | Crop Land | Total Reference Points |
|---|---|---|---|---|---|---|
| Forest Land | 34 | 1 | 1 | 0 | 0 | 36 |
| Settlements | 1 | 30 | 0 | 0 | 0 | 31 |
| Wetland | 0 | 2 | 6 | 1 | 1 | 10 |
| Barren Land | 3 | 2 | 0 | 12 | 2 | 19 |
| Cropland | 1 | 4 | 1 | 1 | 13 | 20 |
| Total Classified Points | 39 | 39 | 8 | 14 | 16 | 116 |

**Table 4.** Error matrix of 5 classes of 2021.

| Classified | Forest Land | Settlements | Wet Land | Barren Land | Crop Land | Total Reference Points |
|---|---|---|---|---|---|---|
| Forest Land | 17 | 0 | 0 | 1 | 0 | 18 |
| Settlements | 1 | 10 | 1 | 0 | 1 | 13 |
| Wetland | 0 | 0 | 3 | 0 | 0 | 3 |
| Barren Land | 0 | 1 | 0 | 2 | 0 | 3 |
| Cropland | 1 | 2 | 0 | 0 | 5 | 8 |
| Total Classified Points | 19 | 13 | 4 | 3 | 6 | 45 |

*2.7. Kappa Analysis*

Kappa analysis was carried out for the agreement of the image classification. Kappa analysis is a discrete multivariate technique used in accuracy assessments. Kappa analysis yields Kappa statistics (an estimate of KAPPA) that is a measure of agreement or accuracy [38]. The Kappa statistic is computed as follows:

$$\text{Kappa co-efficient} = \frac{\text{(Total accuracy-Random accuracy)}}{\text{1-Random accuracy}}$$

$$\text{where, Total accuracy} = \frac{\text{Add all diagonal}}{\text{No. of reference points}}$$

$$\text{Random accuracy} = \frac{\text{Total column} \times \text{Total rows}}{\text{Total ref. points} \cdot \text{Total reference points}}$$

2.7.1. Overall Accuracy

Overall accuracy is a process in which the classified image is determined by dividing the total correct pixels by the total number of pixels in error matrix [39].

$$\text{Overall Accuracy} = \frac{\text{Total number .of correct .pixels}}{\text{Total no. of reference points}}$$

2.7.2. Producer Accuracy

Producer accuracy is a process that estimates how the pixel of land in each category is correctly assigned. Producer accuracy was calculated by following procedure:

$$\text{Producer}' \text{ Accuracy} = \frac{\text{No.of reference sites classified accurately for eachclass}}{\text{Total number of reference sites for each class}}$$

### 2.7.3. User Accuracy

User accuracy was found out by following procedure:

$$\text{User's Accuracy} = \frac{\text{Total no. of correct classification for a particular class}}{\text{Number of row total}}$$

### 2.8. Climatic Variation in Murree (2001–2021)

The climatic data of Murree, obtained from PMD (Pakistan Meteorological Department) Islamabad, was used to assess the climatic variation in Murree for the last 20 years (2001–2021). Table 5 shows the climatic data used for the representation of climatic variation.

**Table 5.** Climatic data of Murree (2001–2021).

| Year | Average Maximum Temperature (°C) | Average Minimum Temperature (°C) | Annual Rainfall (mm) |
|------|----------------------------------|----------------------------------|----------------------|
| 2001 | 18.98 | 7.91 | 1317.10 |
| 2002 | 18.75 | 6.78 | 1264.40 |
| 2003 | 17.73 | 4.65 | 1520.50 |
| 2004 | 18.71 | 6.12 | 1485.10 |
| 2005 | 17.37 | 6.04 | 1616.20 |
| 2006 | 18.68 | 8.32 | 1692.30 |
| 2007 | 18.83 | 9.57 | 1520.60 |
| 2008 | 17.92 | 9.44 | 1593.80 |
| 2009 | 18.28 | 9.91 | 1270.60 |
| 2010 | 18.93 | 9.89 | 1681.60 |
| 2011 | 18.15 | 9.38 | 1442.20 |
| 2012 | 17.58 | 8.81 | 1515.00 |
| 2013 | 18.02 | 9.33 | 1660.10 |
| 2014 | 17.61 | 8.50 | 1561.00 |
| 2015 | 19.00 | 8.96 | 2396.90 |
| 2016 | 19.52 | 10.03 | 1241.20 |
| 2017 | 18.46 | 9.35 | 1335.65 |
| 2018 | 18.72 | 9.51 | 1419.20 |
| 2019 | 17.12 | 8.53 | 1623.32 |
| 2020 | 17.85 | 9.13 | 1592.00 |
| 2021 | 19.24 | 9.62 | 1177.30 |

### 2.9. Geostatistical Analysis

Climatic data of Murree was mapped. The Empirical Bayesian Kriging Interpolation technique was used [40] for the interpolation of climatic data. The ArcGIS geo-statistical analyst module was used to execute it. Tables 6–8 show the climatic data of four Met. Station, for the years 2001, 2011 and 2021 respectively, were used for interpolation.

**Table 6.** Climatic data of four Met. Station for the year 2001.

| Met Station | X | Y | Year | Max. Mean Annual Temp (°C) | Min. Mean Annual Temp (°C) | Annual Rainfall (mm) |
|-------------|------|-------|------|------|------|--------|
| Murree | 73.38 | 33.9 | 2001 | 18.98 | 7.91 | 1317.1 |
| Rawalpindi | 73.21 | 33.4 | 2001 | 30.23 | 15.43 | 1177.71 |
| Islamabad | 73.1 | 33.62 | 2001 | 30.27 | 14.15 | 1472.11 |
| Jehlum | 73.73 | 32.93 | 2001 | 31.61 | 16.91 | 746.72 |

**Table 7.** Climatic data of four Met. Station for the year 2011.

| Met Station | X | Y | Year | Max. Mean Annual Temp (°C) | Min. Mean Annual Temp (°C) | Annual Rainfall (mm) |
|---|---|---|---|---|---|---|
| Murree | 73.38 | 33.9 | 2011 | 18.15 | 9.383333333 | 1442.20 |
| Rawalpindi | 73.21 | 33.4 | 2011 | 28.93333333 | 15.925 | 1254.01 |
| Islamabad | 73.1 | 33.62 | 2011 | 28.93333333 | 14.4 | 1079.61 |
| Jehlum | 73.73 | 32.93 | 2011 | 30.78333333 | 16.95833333 | 748.32 |

**Table 8.** Climatic data of four Met. Station for the year 2021.

| Met Station | X | Y | Year | Max. Mean Annual Temp (°C) | Min. Mean Annual Temp (°C) | Annual Rainfall (mm) |
|---|---|---|---|---|---|---|
| Murree | 73.38 | 33.9 | 2021 | 19.20 | 9.6 | 1177.30 |
| Rawalpindi | 73.21 | 33.4 | 2021 | 28.02 | 16.03 | 1013.31 |
| Islamabad | 73.1 | 33.62 | 2021 | 27.97 | 14.11 | 1150.12 |
| Jehlum | 73.73 | 32.93 | 2021 | 30.12 | 17 | 837.09 |

*2.10. Correlation between Forest Cover Change and Climate Variation*

Data of forest cover change obtained from LULC classified map and climatic data of the study area were correlated and represented graphically.

Graphical Representation

Murree climatic data and forest cover area (2001–2021) were represented graphically. Table 9 is showing the climatic data of Murree and forest cover area for the last 20 years (2001–2021).

**Table 9.** Average maximum, minimum temperature and rainfall data of Murree and forest cover area. for the last 20 years (2001–2021).

| Year | Average Max. Temp.(°C) | Average Min. Temp.(°C) | Annual Rainfall (mm) | Forest Cover Area (ha) |
|---|---|---|---|---|
| 2001 | 18.98 | 7.91 | 1317.10 | 21,411.06 |
| 2002 | 18.75 | 6.78 | 1264.40 | |
| 2003 | 17.73 | 4.65 | 1520.50 | |
| 2004 | 18.71 | 6.12 | 1485.10 | |
| 2005 | 17.37 | 6.04 | 1616.20 | |
| 2006 | 18.68 | 8.32 | 1692.30 | |
| 2007 | 18.83 | 9.57 | 1520.60 | |
| 2008 | 17.92 | 9.44 | 1593.80 | |
| 2009 | 18.28 | 9.91 | 1270.60 | |
| 2010 | 18.93 | 9.89 | 1681.60 | |
| 2011 | 18.15 | 9.38 | 1442.20 | 18,637.11 |
| 2012 | 17.58 | 8.81 | 1515.00 | |
| 2013 | 18.02 | 9.33 | 1660.10 | |
| 2014 | 17.61 | 8.50 | 1561.00 | |
| 2015 | 19.00 | 8.96 | 2396.90 | |
| 2016 | 19.52 | 10.03 | 1241.20 | |
| 2017 | 18.46 | 9.35 | 1335.65 | |
| 2018 | 18.72 | 9.51 | 1419.20 | |
| 2019 | 17.12 | 8.53 | 1623.32 | |
| 2020 | 17.85 | 9.13 | 1592.00 | |
| 2021 | 19.24 | 9.62 | 1177.30 | 17,907.48 |

### 2.11. Questionnaire Data Analysis

In this research, quantitative research method was used. Therefore, collected data were analyzed through frequencies and percentage. Statistical packaged for social science (SPSS) was used as an efficient tool for data analysis.

## 3. Results

### 3.1. LULC Classification

In this research, three land-use and land-cover maps were created through supervised classification of the study area. The following are the five LULC classes that were identified after classification; forest, built-up area/settlements, croplands, barren land and water.

### 3.2. Forest Cover Analysis 2001

Through supervised classification of the 2001 Landsat 5 TM image, 5 LULC classes were obtained as shown in Figure 4. These classifications were estimated in hectares (ha) as well as percentages, depending on the number of pixels counted (percent).

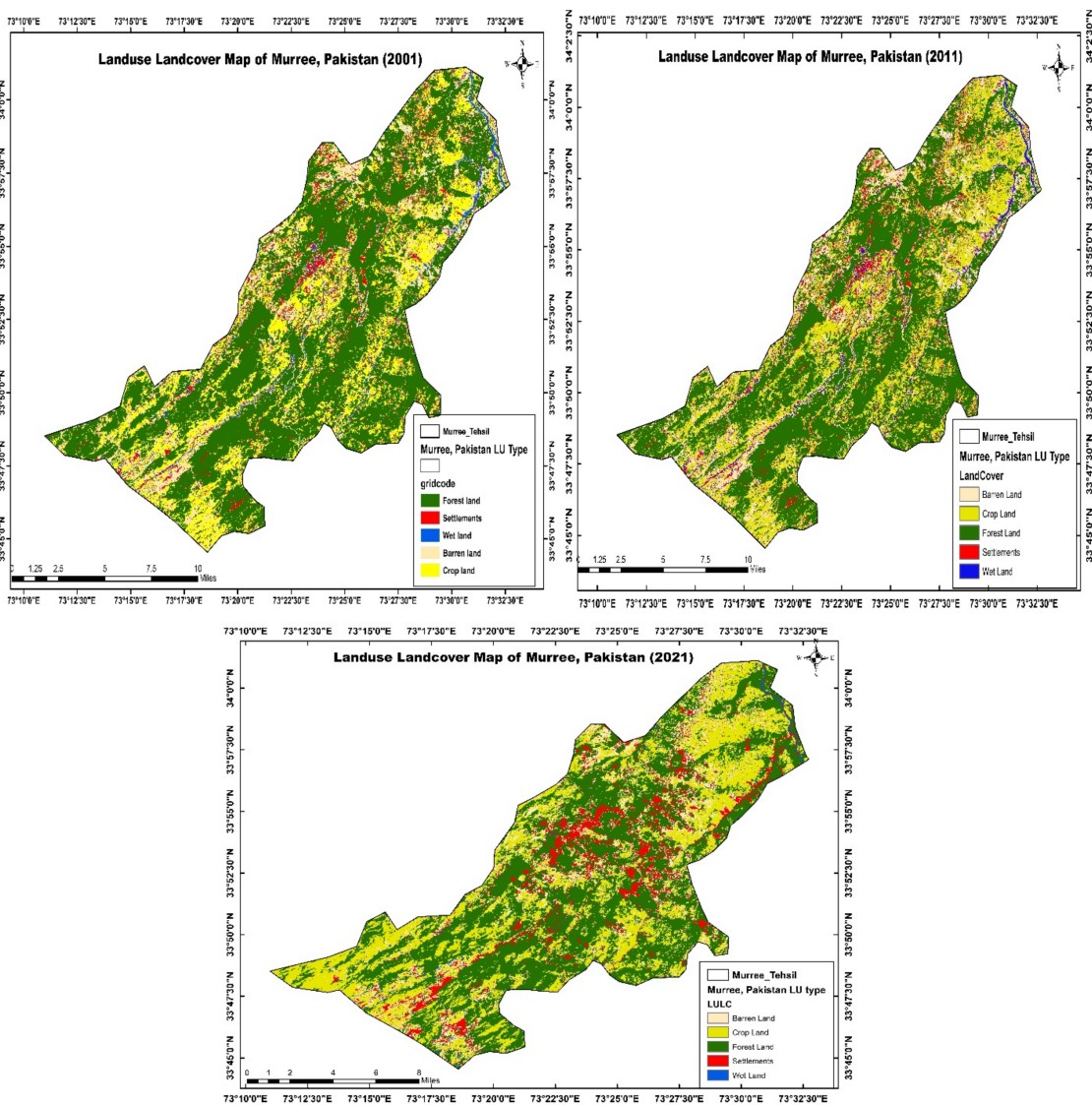

**Figure 4.** LULC map of Murree, showing five land-use land-cover (LULC) classes (2001, 2011 and 2021).

Out of the five categories of LULCC, the forest was 55.98% of the region, covering 21,411 ha of area, followed by cropland (8117 ha) with 21.22 percent of the area. The cropland class is followed by barren land (5521 ha) with 14.43 percent of the total area, then the settlement class (2243 ha) with 5.86 percent, while wetland (956 ha), covering 2.5 percent of the total area, was the LULCC with the smallest area.

### 3.3. Forest Cover Analysis 2011

The five LULC classes shown in Figure 4 were obtained through supervised classification of the 2011 Landsat 5 TM image. These classifications were estimated in hectares (ha) as well as percentages, depending on the number of pixels counted (percent).

Out of the five categories of LULC, the largest area covered by forest was 50.63 percent (18,637 ha) of the region. The cropland (9260 ha) came in second with 25.16 percent. It was followed by barren land (5582 ha) with 15.16 percent, then settlements (2368 ha) with 6.43 percent, while water bodies (963 ha), covering 2.62 percent of the total area, was the LULC with the smallest area.

### 3.4. Forest Cover Analysis 2021

The five LULC classes shown in Figure 4 were obtained through supervised classification of the 2021 Landsat 8 OLI image. These classifications were estimated in hectares (ha) as well as percentages, depending on the number of pixels counted (percent).

Out of the five categories of LULC, the largest area covered by forest was 47.72 percent (17,907.48 ha) of the region. The cropland (11,588.04 ha) came in second with 30.87 percent. It is followed by barren land (4549.05 ha) with 12.12 percent, then settlements (3074.04 ha) with 8.19 percent, while water body (407.52 ha), covering 1.08 percent of the total area, was the LULC with the smallest area.

### 3.5. Accuracy Assessment 2001

An important step in image classification is the accuracy evaluation of the classified image. Accuracy determines how effective a thematic map made from a satellite image is. A LULC accuracy assessment was performed for the Landsat image of 2001 and an analytical accuracy report was produced from the error matrix as shown in Table 3. For forest land, user accuracy was 87.5 percent. The user accuracy for barren land was 80 percent, for settlements it was 71.43 percent, while for crop land and wet land it was 62.5 percent for both (shown in Table 10). Producer accuracy for crop land was 100 percent, and for settlements it was 71.43 percent. Forest land and barren land had producer accuracy of 70 percent and 66.67 percent, respectively, whereas wet land had a producer accuracy of 62.5 percent (shown in Table 10). The overall accuracy of the classification of the 2001 image was 72.09%, as shown in Table 10. The Kappa co-efficient of the 2001 LULC classified map was 0.72.

**Table 10.** User's, producer's accuracies and overall accuracy of the classified imagery of 2001.

| User's Accuracy | Percentage | Producer's Accuracy | Percentage | Overall Accuracy | |
|---|---|---|---|---|---|
| Forest Land | 87.5 | Forest Land | 70 | Total Correct Reference Points | 43 |
| Settlements | 71.43 | Settlements | 71.43 | Total "True" Reference Points | 31 |
| Wetland | 62.5 | Wetland | 62.5 | Overall Accuracy | 72.09% |
| Barren Land | 80 | Barren Land | 66.67 | | |
| Cropland | 62.5 | Cropland | 100 | | |

### 3.6. Accuracy Assessment 2011

For the Landsat image of 2011, a LULC accuracy assessment was performed, and an analytical accuracy report was produced from the error matrix as shown in Table 4. User accuracy for forest land was 94.44 percent. Settlements and cropland's user accuracy were 96.77 percent and 65 percent, respectively, while barren land's user accuracy was 63.16 percent, with wet land accounting for 60 percent of user accuracy (shown in Table 11). Forest land's producer accuracy was 87.18 percent, followed by barren land with 85.71 percent producer accuracy. Producer accuracy of crop land and settlements were 81.25 percent and 76.92 percent respectively, while wet land showed 75 percent producer accuracy (shown in Table 11). The overall accuracy of the classification of 2011 images was 81.89%, as shown in Table 11. The Kappa co-efficient of 2011 LULC classified map was 0.81.

**Table 11.** User's, producer's accuracies and overall accuracy of the classified imagery of 2011.

| User's Accuracy | Percentage | Producer's Accuracy | Percentage | Overall Accuracy | |
| --- | --- | --- | --- | --- | --- |
| Forest Land | 94.44 | Forest Land | 87.18 | Total Correct Reference Points | 116 |
| Settlements | 96.77 | Settlements | 76.92 | Total "True" Reference Points | 95 |
| Wetland | 60 | Wetland | 75 | Overall Accuracy | 81.89% |
| Barren Land | 63.16 | Barren Land | 85.71 | | |
| Cropland | 65 | Cropland | 81.25 | | |

### 3.7. Accuracy Assessment 2021

For the Landsat image of 2021, a LULC accuracy assessment was performed, and an analytical accuracy report was produced from the error matrix as shown in Table 5. Wetland accounts for 100 percent user accuracy. User accuracy for forest land was 94.44 percent. Settlements' and barren land's user accuracy were 76.92 percent and 66.67 percent, respectively, and cropland's user accuracy was 62.5 percent (shown in Table 12). Forest land's producer accuracy was 89.47 percent, followed by cropland with 83.33 percent producer accuracy. Producer accuracy of settlements and wetland were 76.92 percent and 75 percent respectively, while barren land shown 66.67 percent producer accuracy (shown in Table 12). The overall accuracy of the classification of the 2011 image was 82.22%, as shown above in Table 12. The Kappa co-efficient of the 2021 LULC classified map was 0.82.

**Table 12.** User's and producer's accuracies of the classified imagery of 2021.

| User's Accuracy | Percentage | Producer's Accuracy | Percentage | Overall Accuracy | |
| --- | --- | --- | --- | --- | --- |
| Forest Land | 94.44 | Forest Land | 89.47 | Total Correct Reference Points | 45 |
| Settlements | 76.92 | Settlements | 76.92 | Total "True" Reference Points | 37 |
| Wet Land | 100 | Wet Land | 75 | Overall Accuracy | 82.22% |
| Barren Land | 66.67 | Barren Land | 66.67 | | |
| Crop Land | 62.5 | Crop Land | 83.33 | | |

*3.8. Climatic Variation in Murree (2001–2021)*

During the time period from 2001 to 2021, there were noticeable fluctuations in the mean minimum temperature, mean maximum temperature and precipitation in Murree.

3.8.1. Average Annual Minimum Temperature (2001–2021)

Murree's average annual minimum temperature was 7.91 °C in 2001 and increased to 9.38 °C in 2011 and then risen from 9.38 °C to 9.62 °C in 2021, as shown in Figure 5. According to the findings of the climatic data of Murree, the average annual minimum temperature fluctuated (increased) by 1.71 °C throughout the 20 years (2001–2021). The distribution of average annual minimum temperature in Murree, for the years 2001, 2011 and 2021, was also mapped using GIS technique for better understanding of climatic variation, as shown in Figure 5.

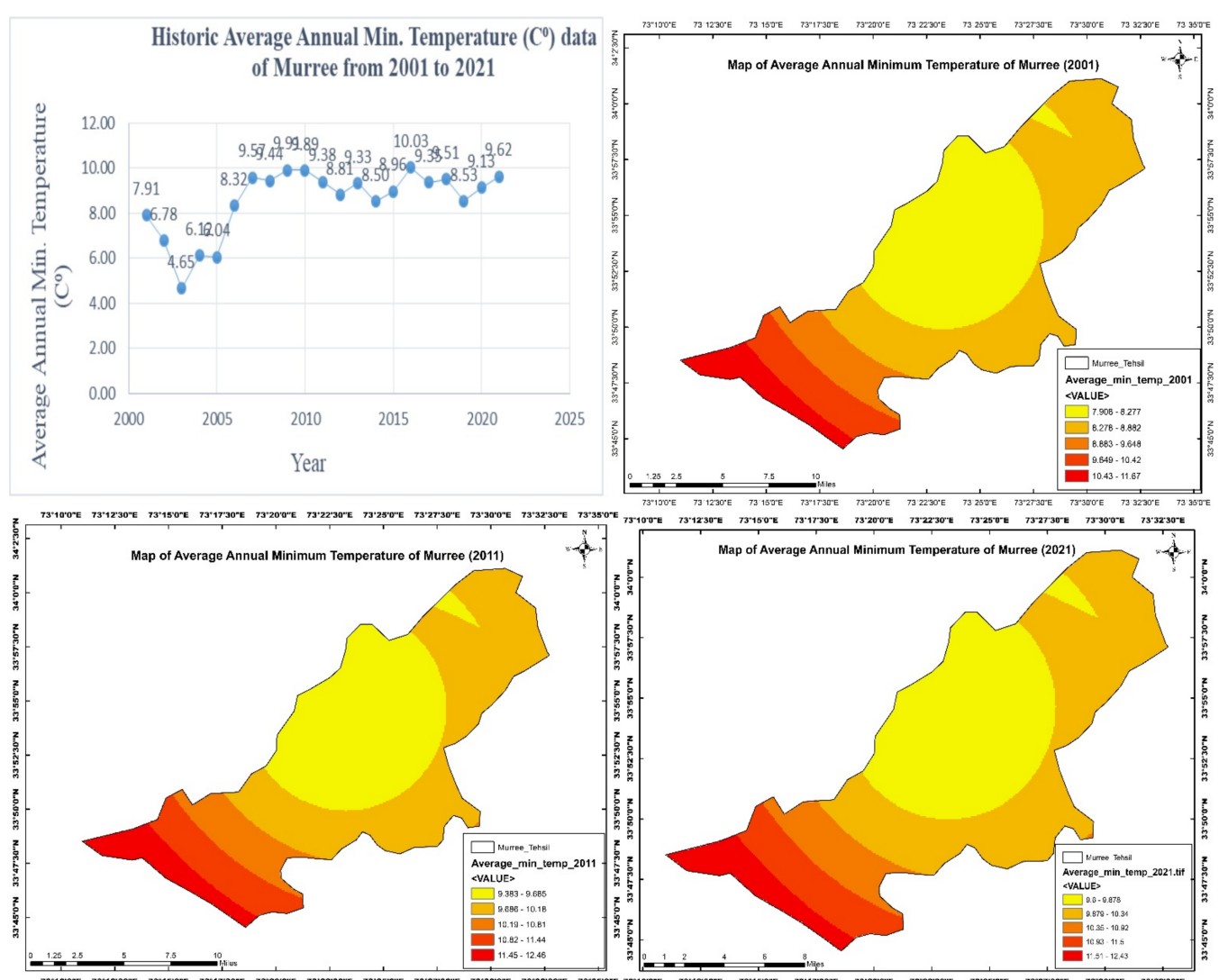

**Figure 5.** Average annual minimum temperature (°C) data and temperature distribution of Murree (from 2001–2021).

3.8.2. Average Annual Maximum Temperature (2001–2021)

The average annual maximum temperature of Murree was 18.98 °C in 2001, fluctuating between 18.98 °C in 2001 and 18.15 °C in 2011, then increasing from 18.15 °C in 2011 to 19.24 °C in 2021 (shown in Figure 6). According to the findings, Murree's average annual maximum temperature has risen 0.26 °C in the past 20 years. The distribution of average

annual maximum temperature in Murree, for the year 2001, 2011 and 2021, was also mapped using GIS technique for better understanding of climatic variation, as shown in Figure 6.

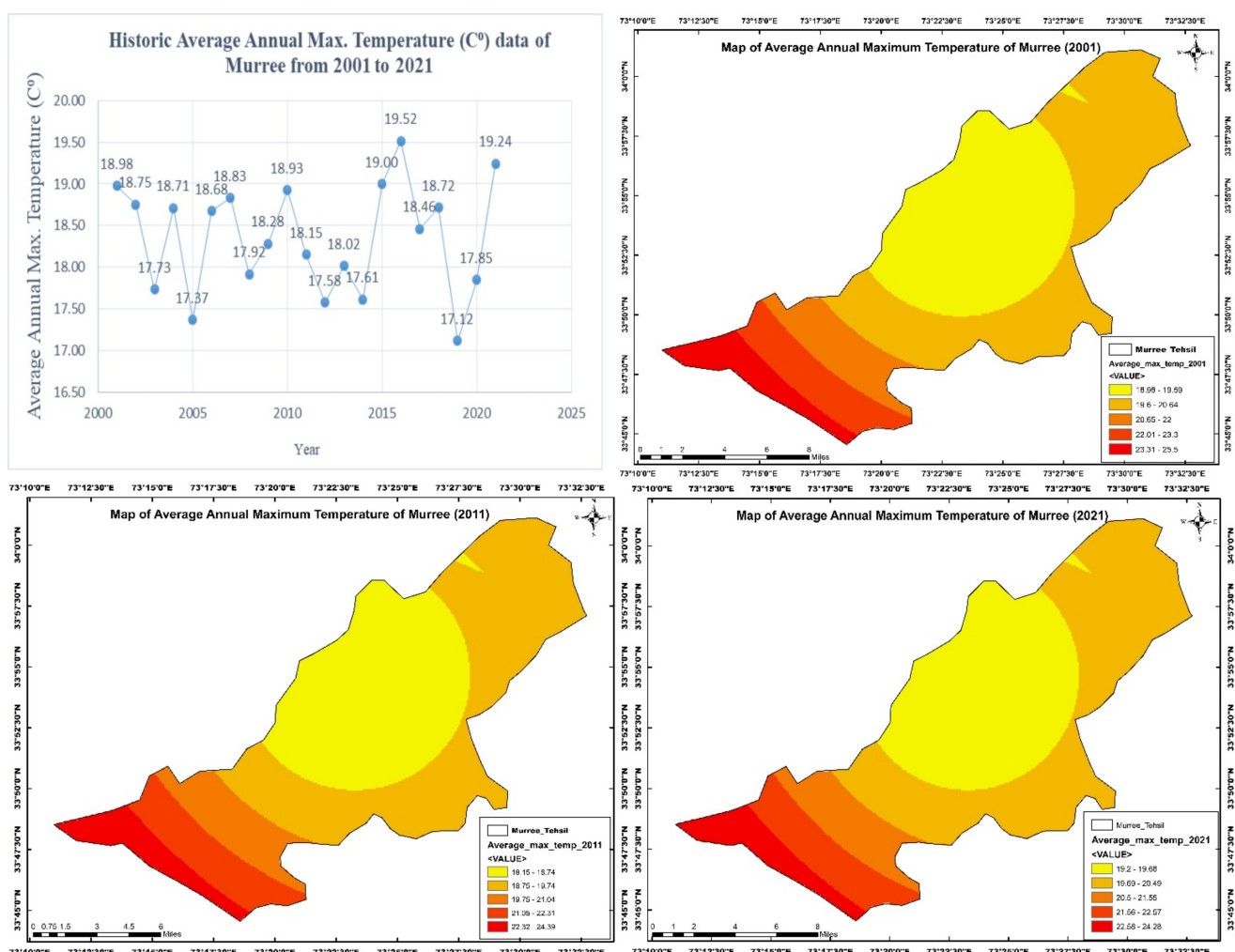

**Figure 6.** Average annual max. temperature (°C) data and temperature distribution of Murree from 2001–2021.

### 3.8.3. Annual Precipitation (2001–2021)

Murree's annual precipitation was 1317.10 mm in 2001. Precipitation fluctuated from 1317.10 mm in 2001 to 1442.20 mm in 2011 and then declined to 1177.30 mm in 2021. The annual precipitation in Murree has declined over the past 20 years, as shown in Figure 7. The distribution of annual precipitation in Murree, for the year 2001, 2011 and 2021, was also mapped using GIS technique for better understanding of climatic variation, as shown in Figure 7.

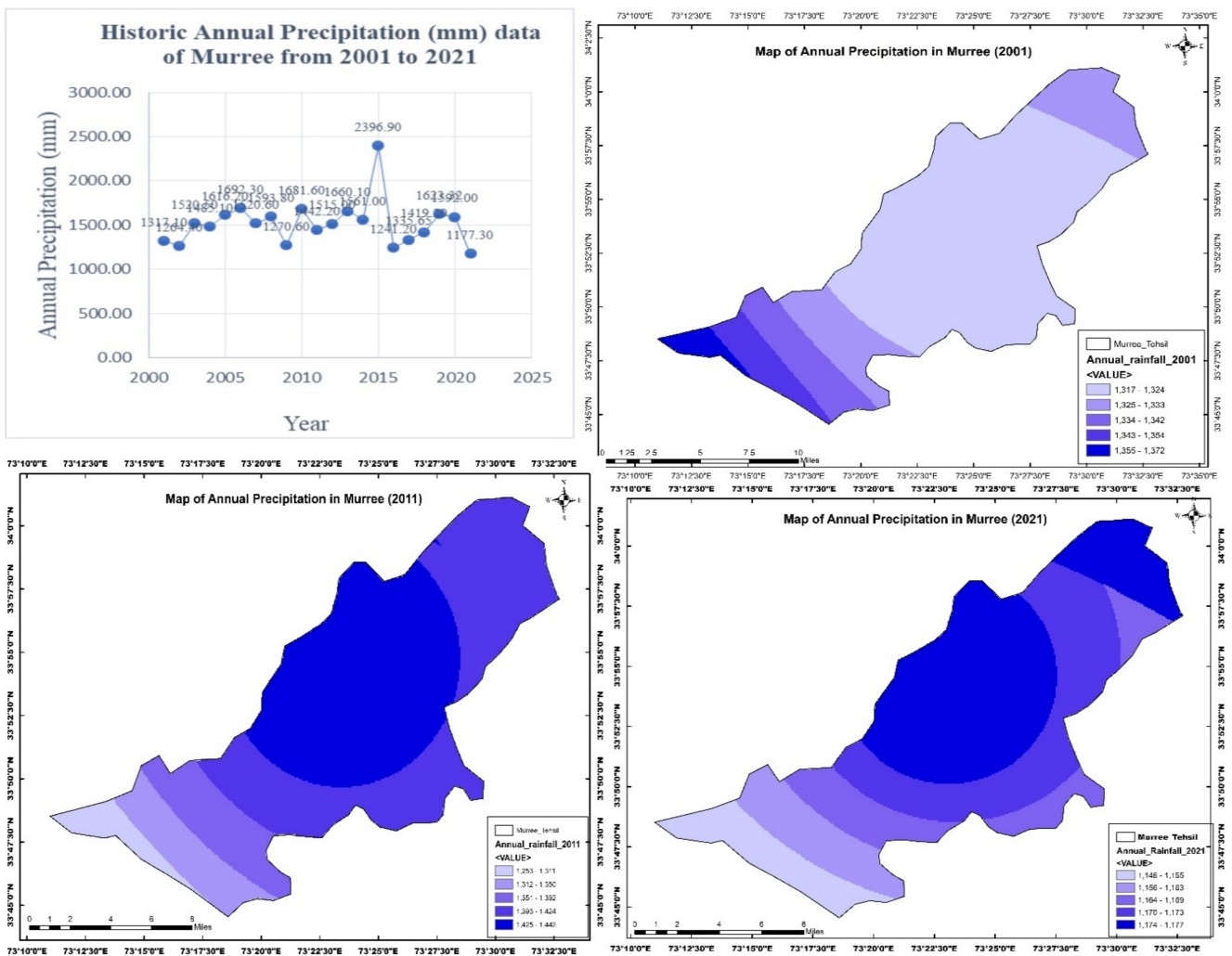

**Figure 7.** Annual precipitation (mm) data and precipitation distribution of Murree from 2001–2021.

### 3.9. Correlation between Forest Cover Change and Climate Variation

3.9.1. Average Maximum and Minimum Temperature and Forest Cover (2001–2021)

In Murree, significant changes in forest cover were seen and showed a decreasing trend during 2001–2021. Forest cover was 21,411.06 ha in 2001. In 2011, it shrank to 18637.11 ha and in 2021 the forest cover further decreased to 17,907.48 ha (shown in Figure 8). Notable changes were also recorded in the average maximum and minimum temperatures in Murree since the last 20 years. In 2001, the average maximum temperature was recorded at 18.98 °C. Then it fluctuated to 18.15 °C in 2011 and 19.24 °C in 2021 as shown in Figure 8. In 2001 the average minimum temperature of Murree was recorded 7.91°C, which fluctuated (increased) to 9.38 °C in 2011 and it further increased to 9.62 °C in 2021 as shown in Figure 8.

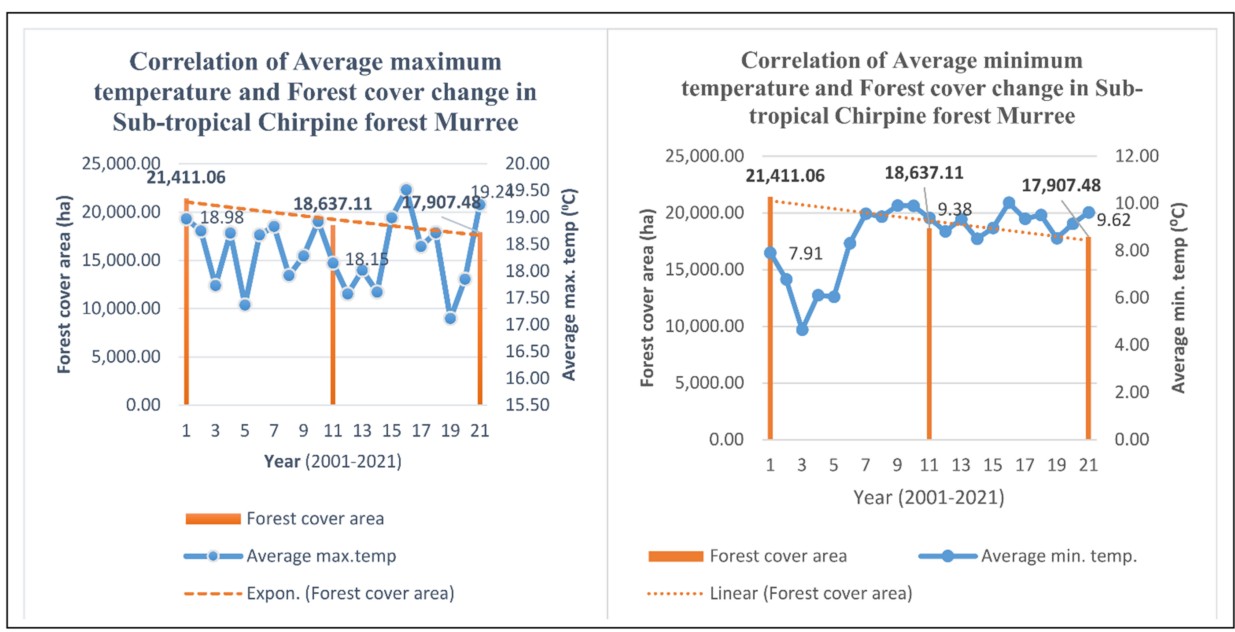

**Figure 8.** Graphical representation of correlation of average maximum and minimum temperature and forest cover change in subtropical Chir pine forest, Murree.

### 3.9.2. Annual Rainfall and Forest Cover (2001–2021)

According to Figure 9, annual rainfall in Murree was 1317.10 mm in 2001 when the forest cover was 21,411.06 ha. With the passage of time, the annual rainfall has shown rise and fall and in 2011, when forest cover was 18,637.11 ha, the annual rainfall was recorded at 1442.20 mm in Murree. Then from 2011 to 2021, it showed fluctuation and decreased to 1177.30 mm in 2021 with a forest cover of 17,907.48 ha.

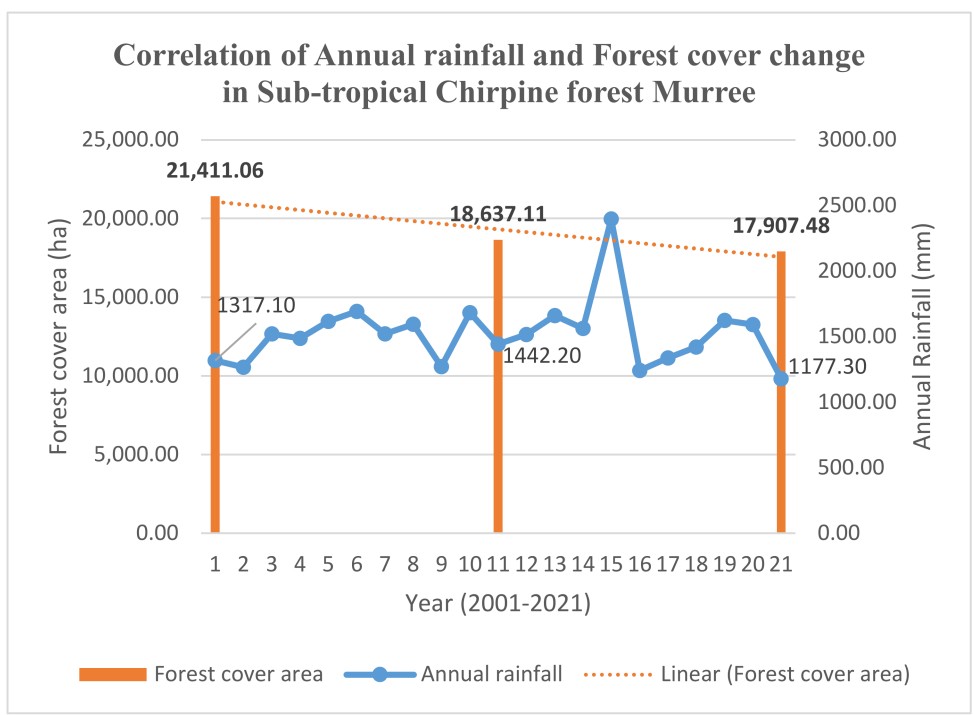

**Figure 9.** Graphical representation of correlation of annual rainfall and forest cover change in subtropical Chir pine forest, Murree.

*3.10. Drivers of Deforestation*

3.10.1. Socio-Economic Characteristics

Gender, Age and Marital Status Distribution

In this research, all the questions were asked to males, because females were reluctant to respond due to cultural issues. The majority of respondents were of the ages between 18 and 25 years (42%), followed by respondents of ages of above 46 years (26%), while others were of the ages between 26 and 35 years (20%) and 36–45 years (12%). Most of the respondents were married (54%), while others were unmarried (46%).

Education and Occupation

Most of the respondents had qualifications of intermediate (38%) and graduate (26%). In total, 20% of the respondents had matric qualification and 14% of the respondent were primary passed, while 2% of the respondent were uneducated. The data revealed that 90% of the respondents had either their own business or were students, while 5% of the respondents were government job holders.

3.10.2. Drivers of Deforestation

Rate of Deforestation

Table 13 illustrates that 60% of the total respondents said that the rate of deforestation is slow, 22% said that it is rapid and 18% said that the rate is moderate. Analysis of the data revealed that the overall rate of deforestation in the study area is slow, as 60% of the respondents said that it is slow.

**Table 13.** Rate of deforestation in Tehsil Murree.

| Rate of Deforestation | Frequency | Percent |
|:---:|:---:|:---:|
| Rapid | 11 | 22.0 |
| Slow | 30 | 60.0 |
| Moderate | 9 | 18.0 |
| Total | 50 | 100.0 |

People Dependency on Forest for Living

Data collected from the respondents illustrated that the people of the area do not depend on the forest for their livelihood, as 80% of the total respondents said they do not rely on the forest for their living and only 20% responded that they earn from the forest for their livelihood.

Source of Anthropogenic Deforestation

According to data collected from the survey, social use of forest is the main source of anthropogenic deforestation in the study area, as 58% of the total respondents said, followed by economic use (30% of the total respondents responded), while demographic (6%) and other uses (6%) are equally responsible for anthropogenic deforestation.

Main Drivers of Deforestation

Most of the people (54%) were of the opinion that fuelwood for use is the main driver of deforestation in the study area. In total, 16% of the respondents said that timber production is the main driver of deforestation, while 8% of people said that urbanization is responsible for deforestation. In total, 22% of the people identified other drivers, e.g., agriculture expansion, as the drivers of deforestation in the study area (shown in Table 14). The data revealed that people are mostly cutting forests for social use, as mentioned above.

**Table 14.** Main drivers of deforestation in Murree.

| Main Drivers of Deforestation | Frequency | Percent |
|---|---|---|
| Fuelwood production | 27 | 54.0 |
| Timber production | 4 | 8.0 |
| Urbanization | 8 | 16.0 |
| Others | 11 | 22.0 |

Natural Elements Causing Deforestation

The survey indicated that the study area is also facing deforestation due to some natural elements, in which forest fire is the top natural element causing deforestation (88% responded), followed by other elements (10% responded), e.g., erosion, while flooding is also causing deforestation (2% responded).

3.10.3. Impact of Deforestation

Impact of Deforestation on Temperature

Deforestation has an impact on the temperature of the area. In total, 72% of people believed that deforestation has resulted in an increase in the temperature of the area, while 28% of people stated that there has been no change in temperature over the past years. Overall, the majority of the people were of the opinion that temperature of the area has risen over the past years due to the loss of vegetation.

Impact of Deforestation on Rainfall and Snow

Data were collected and analyzed, asking people about the impact of deforestation on rainfall and snowfall. In total, 58% of the whole respondent stated that the rainfall and snow is low as compared to the past and 18% said that rainfall and snowfall has increased, while 24% believed that there is no change in the rate of rainfall and snow over the past years. The data revealed that deforestation has resulted in a lower rate of rainfall and snowfall.

Chi-Square Test

A chi-square test was also applied to check the effect of deforestation on the climate. Analysis showed that the relation between deforestation and climate variation is significant as the *p*-value, i.e., 0.04, is less than 0.05 as shown in Table 15.

**Table 15.** Deforestation effects on climate.

| Deforestation Effects on Climate | Agreed | Not Agreed | Df | $X^2$ | $\alpha$ | *p*-Value |
|---|---|---|---|---|---|---|
| Increase in temperature due to deforestation | 36 | 14 | | | | |
| Increase in rainfall due to deforestation | 9 | 12 | 2 | 6.23 | 0.05 | 0.04 |
| Decrease in rainfall due to deforestation | 29 | 12 | | | | |

Deforestation and Erosion

The study area has been affected by erosion due to the cutting of trees, as 78% of the respondents said that deforestation has caused erosion. Only 22% of people did not agree with the above statement.

### 3.10.4. Measures to Reduce Deforestation

Engagement in Forest Conservation Activities

Local people were asked about their involvement in forest conservation activities and 72% of people responded that they were not engaged in any forest conservation activity, showing a lack of awareness of the local inhabitants towards the importance of trees and forests.

Convincing People to Stop Cutting of Trees

When respondents were asked about their role in convincing people to stop cutting trees, 86% responded that they were not involved in such activities to reduce deforestation.

## 4. Discussion

Forest cover change analysis revealed that in 2001 the forest area of Murree was 55.98%, which reduced to 50.63% in 2011 and in 2021 it further reduced to 47.72%. These results were in line with Saeed et al. [41]. They stated that the forest area of Murree was 55% in 1999 and it was reduced to 49% in 2011. The results of forest cover change classification of the last 20 years (2001–2021) of Murree showed that there was a decrease in the forest cover area of Murree by 8.26%. These results were similar to Kausar et al. [42]. They detected LULC change for the years 1998, 2003, 2005 and 2010 using GIS and stated that the forests of Murree have decreased by 12.7%. A similar study was conducted by Amjad et al. [8] in Mansehra. They assessed forest cover in Mansehra from 1998 to 2017, using GIS and remote sensing. Results of this study showed that the forest cover was 14 percent (601 square kilometers) in 1998, 15 percent (668 square kilometers) in 2008, and in 2017, it was 5 percent (194 square kilometers), which indicated a downward trend in forest cover area, similar to the current study.

The correlation between forest cover change and climate variations showed that as the forest cover decreased from 21,411.06 ha in 2001 to 17,907.48 ha in 2021, the average maximum temperature has risen 0.26 °C and the average minimum temperature has risen 1.710 °C in the past 20 years (2001–2021), whereas the annual rainfall has decreased 139.8 mm during the time span of 2001–2021, which shows that deforestation is causing an increase in temperature and a decrease in rainfall. The same study was carried out by Anjali & Roshni [43] in Kerala, India. They used GIS and remote sensing techniques to detect forest cover change and correlate it with rainfall and temperature in the region during the time period from 2000 to 2019. The results stated that forest area has decreased by 12.65%, whereas minimum and maximum land surface temperatures have increased by 0.9 °C and 5 °C, respectively. Furthermore, it was observed that with the decline of forest cover, the rainfall was also decreased, but the rainfall decline was not significantly noted. Shakun et al. [9] also stated that deforestation is the main factor contributing to the decrease in rainfall and increase in temperature globally.

Questionnaire survey data revealed that most of the people (54%) were of the opinion that fuelwood for domestic use is the main driver of deforestation in the study area. This result was similar to Ullah et al. [44], as they carried out a questionnaire survey within Teknaf Wildlife Sanctuary, Bangladesh and revealed that 37% of the households in that area are involved in fuelwood collection. In total, 22% of the people identified other drivers, i.e., agriculture expansion, as the drivers of deforestation in Murree. This result was in line with Sajjad et al. [36], as they noted agriculture expansion as the main driver of deforestation in Tehsil Barawal, Dir Upper, Pakistan. In total, 16% of the respondents said timber production is the responsible factor for deforestation in the study area. Ali et al. [45] conducted a study, consistent with this finding, in Basho valley, Skardu and stated that the main causes of deforestation in Basho Valley are inefficient administration and unauthorized timber harvesting for commercial use, supported by the Forest Department. In total, 8% of people said that urbanization is responsible for deforestation, relevant to the findings of Shukla et al. [46], as they examined urbanization and unchecked infrastructure as drivers of deforestation in a river basin.

## 5. Conclusions

In this study, we detected forest cover change and correlated it with the climatic variables (minimum and maximum temperature and precipitation) in the subtropical Chir pine forest, Murree, Pakistan. This research also identified the main drivers of deforestation in the study area. Five (5) land-use land-cover (LULC) categories were demarcated to detect a change from 2001 to 2021. Using ArcMap version 10.5, the supervised maximum likelihood classification (MLC) technique was applied to satellite imageries for classification. Climatic data was interpolated by empirical Bayesian kriging interpolation and it was correlated with forest cover change graphically. Drivers of deforestation were identified through questionnaires.

It has been noted that as the forest area decreased by 8.26% from 2001 to 2021, the average maximum temperature has risen 0.26 °C and the average minimum temperature has risen 1.710.26 °C in the past 20 years. The annual rainfall in Murree has also decreased by 139.8 mm during the time span of 2001–2021, showing that forest decline has caused an increase in temperature and a decrease in rainfall in Murree. Fuelwood (54%), agriculture expansion (22%), timber production (16%) and urbanization (8%) were recorded as drivers of deforestation in the study area.

Based on the findings, it has been concluded that deforestation can impact the climate variables of a region. Therefore, there is a need to monitor forest cover after 5–10 years in the current climate-changing scenario. It is also recommended to provide gas facilities to stop cutting trees for fuelwood and control illicit cutting by the timber mafia. An awareness campaign about the importance of community participation in forest conservation is necessary to conserve these valuable resources.

**Author Contributions:** Methodology, L.A.; formal analysis, H.T.; investigation, M.I., K.M. and M.M.; resources, I.H.; data curation, I.H.; writing—original draft preparation, W.A.; supervision, L.A.; project administration, A.S. All authors have read and agreed to the published version of the manuscript.

**Funding:** This research received no external funding.

**Data Availability Statement:** All the data already provided.

**Conflicts of Interest:** The authors declare no conflict of interest.

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
