# Peer review of "Forest Cover Change and Climate Variation in Subtropical Chir Pine Forests of Murree through GIS"

_forests, doi:10.3390/f13101576_

Round 1

Reviewer 1 Report

In general, I believe the research will be of interest to some readers. The methodology used in the research is appropriate, I find it scientifically appropriate.

1- Figures should be rearranged, because the numbers are mixed in some places.

2-Some images should be improved,

3- I have shown few corrections on the text.

Author Response

 Response to Reviewer 1 Comments

Point 1: Figures should be rearranged, because the numbers are mixed in some places.

Response 1: Figures have been rearranged and similar images have been adjusted in a single frame.

Point 2:- Some images should be improved,

Response 2: Images are improved now  e.g Study Area Map

Point 3:-  I have shown few corrections on the text

Response 3: Mentioned corrections have been made (Murree, Pakistan)

Reviewer 2 Report

Manuscript "Forest cover change and Climate variation in Subtropical Chir Pine forests of Murree through GIS" detect forest cover change and correlates it with the climatic variables (Minimum & Maximum temperature and precipitation) in Subtropical Chir pine forests, Murree.

Overall, the topic is meaningful and within the journal's scope. The methods used are appropriate and well-chosen for the case. The work done is well structured and well written, with interesting findings. But this work required some revisions and can be considered for publication after a gentle modification.

1.      The authors should arrange the abstract scientifically by presenting the research objectives clearly.

2.      Most of the literature used in this work is too old. Authors should update the information and research gaps with the latest literature.

3.      The 7th paragraph of the introduction. Please have a careful check. It's better to discuss the history of remote sensing data/satellite data availability before referring to RG and GIS's advantages. Later authors mentioned "several studies", but I could find just a reference from 1993.

4.      Figure 1 should be edited' Legends of Figure 1 are unclear. It's better to add the neighbouring borders of Pakistan for a clear understanding of the location. 

5.      The first line of section 2.3. "data was gathered from remotely sensed satellite data (Landsat images)." or download. Also, consider adding the source (USGS website) etc.

6.      In section 2.3 "The authors mentioned, "The reason for this 10-year gap was the SLC-off data, which shows data gaps after May 31, 2003 (Jamal et al., 2018)."

(i). Please consider reporting the sensor name, which contains the SLC error.

(ii). In table 1 authors mentioned that they have utilized "Landsat ETM" Sensor data in this study. Please consider making that clear. Did the author use ETM data, and, at the same time, were they aware of SLC? If so, where is the methodology for processing SCL error fixing or gap filling, etc.?

(iii). Later, in Sections 3.2 and 3.3 Authors reported they used Landsat TM data for 2001 and 2011, which contradicts Table 1.

(iii). Section 3.4. (Figure 10) Are authors sure about the use of Landsat TM data for 2021???

7.      In Table 1 authors mentioned a single row and path, while in section 2.4 authors said they mosaicked two images with path/row 150/036 and 150/037 to cover and extract the whole study area. Please have a careful look and fix this error.

8.      Section 2.3, 2.4, 2.5, 2.6. can be mosaicked and discussed into 2 sections, "Data sets and pre-processing" and "methods", instead of repeating the same work under different headings.

9.      The methodology of this work is so extended that it may cause boring readers. It's better to provide a methodology flowchart for a better and quick understanding.

10.  The conclusion of this work should be extended scientifically. 

11.  It's better to rearrange the decreased figures numbers by adjusting all the similar figures in the same frame instead of a separate caption for each figure.

Author Response

Response to Reviewer 2 Comments

Point 1:The authors should arrange the abstract scientifically by presenting the research objectives clearly.

Response 1: Research objectives are now clearly presented in the Abstract.

Point 2:Most of the literature used in this work is too old. Authors should update the information and research gaps with the latest literature.

Response 2: Literature updated and all the cited literature is from 2001 to 2021, as the research time span is from 2001 to 2021. Research gap has been also updated.

Point 3: The 7th paragraph of the introduction. Please have a careful check. It's better to discuss the history of remote sensing data/satellite data availability before referring to RG and GIS's advantages. Later authors mentioned "several studies", but I could find just a reference from 1993.

Response 3: In the 7th paragraph, RS and GIS uses has been discussed. Several studies has been cited with several references.

Point 4: Figure 1 should be edited' Legends of Figure 1 are unclear. It's better to add the neighbouring borders of Pakistan for a clear understanding of the location. 

Response 4: Study area map has been updated.

Point5: The first line of section 2.3. "data was gathered from remotely sensed satellite data (Landsat images)." or download. Also, consider adding the source (USGS website) etc.

Response 5: Referring to section 2.3, data was gathered from USGS website.

Point6: In section 2.3 "The authors mentioned, "The reason for this 10-year gap was the SLC-off data, which shows data gaps after May 31, 2003 (Jamal et al., 2018)."

Response 6:

i) Sensor name was Landsat 7 ETM.

ii) Landsat 5 TM data was used for 2001 and 2011, while Landsat 8 OLI data was used for 2021.

iii) Fixed according to Table 1

iv) Landsat 8 OLI was used for 2021, fixed according to Table 1. 

Point 7: In Table 1 authors mentioned a single row and path, while in section 2.4 authors said they mosaicked two images with path/row 150/036 and 150/037 to cover and extract the whole study area. Please have a careful look and fix this error.

Response 7: Two images of path and row 150/036 and 150/037 was used to mosaicked 2011 image 

Point 8: Section 2.3, 2.4, 2.5, 2.6. can be mosaicked and discussed into 2 sections, "Data sets and pre-processing" and "methods", instead of repeating the same work under different headings.

Response8: Section 2.3, 2.4, 2.5, 2.6 has been mosaicked and discussed in 2 sections.

Point9: The methodology of this work is so extended that it may cause boring readers. It's better to provide a methodology flowchart for a better and quick understanding.

Response 9: Methodology framework has been provided for better understanding.

Point 10: The conclusion of this work should be extended scientifically. 

Response 10: The conclusion has been extended scientifically.

Point 11:  It's better to rearrange the decreased figures numbers by adjusting all the similar figures in the same frame instead of a separate caption for each figure.

Response 11: Figures have been rearranged and similar images have been adjusted in a single frame.

Round 2

Reviewer 2 Report

Manuscript has been significantly improved and can be accepted for publication.

Author Response

Point 1: Definition of forest is not documented; Pls, clarify how LU-class forest is determined; according to canopy cover and size area ??

Response 1: A land area of more than or at least 0.5 hectare area, with trees of  more than 5 meter height and a conopy cover of more than 10 % (FAO).

LU forest area is determined according to canopy cover. More than 10% canopy cover has been classified as forest.

Point 2: How many trees are accepted in croplands and settlements???

Response 2: Less than 10% canopy cover has been declared non-forested area, and other land-use categories have been determined according to majority feature. For example if any area has more settlements than the area has been classified as settlements.

Point 3: Discussion could have few sentences about uncertainty of catergories!!

Response 3: In Discussion, only Forest land class has been discussed with the previous studies, as our main concern is forest.
